# Why Does Monk Fruit Extract Remain Only Partially Approved in the EU? Regulatory Barriers and Policy Implications for Food Innovation

**DOI:** 10.3390/foods14162810

**Published:** 2025-08-13

**Authors:** Urszula Kaim, Urszula Gawlik, Katarzyna Lisiecka

**Affiliations:** 1Department of Bioprocess Engineering, Wrocław University of Economics & Business, St. Komandorska 118-120, 53-345 Wrocław, Poland; 2Department of Biochemistry and Food Chemistry, University of Life Sciences in Lublin, St. Akademicka 13, 20-950 Lublin, Poland; urszula.gawlik@up.lublin.pl (U.G.); katarzyna.lisiecka@up.lublin.pl (K.L.)

**Keywords:** monk fruit extract, non-caloric sweeteners, European Food Safety Authority (EFSA), food regulation, novel foods, functional foods, regulatory barriers, sustainable food systems

## Abstract

Monk fruit extract (*Siraitia grosvenorii*, MFE) is a natural, non-caloric sweetener known for its intense sweetness, antioxidant properties, and potential metabolic health benefits. While certain aqueous monk fruit decoctions are recognised as non-novel foods in the UK and Ireland due to significant pre-1997 consumption, the European Union (EU) has adopted a more cautious approach under the Novel Food Regulation (EU) 2015/2283. As of October 2024, only one specific aqueous extract of monk fruit has been authorised in the EU under Regulation (EU) 2024/2345, permitting its use in several food categories. However, highly purified mogrosides and non-aqueous extracts remain unapproved due to gaps in toxicological data and the absence of industry-led applications. This review systematically analyses the EU’s regulatory barriers, comparing MFE’s legal status with other approved non-caloric sweeteners such as steviol glycosides and erythritol, and examining regulatory frameworks in the EU, United States, United Kingdom, and China. Findings indicate that although 18 non-caloric sweeteners are currently authorised in the EU, regulatory constraints continue to hinder the broader approval of MFE, limiting innovation and the availability of natural sweeteners for consumers. Harmonising regulations, leveraging international safety assessments, and promoting industry engagement are recommended to advance MFE’s authorisation and support sustainable food innovation in the EU. Addressing these challenges is crucial to ensure that European consumers and industry can benefit from safe, innovative, and health-promoting alternatives to sugar, aligning food policy with broader public health goals and sustainability commitments.

## 1. Introduction

The global rise in metabolic disorders, including obesity, type 2 diabetes, and cardiovascular diseases, has intensified efforts to reduce added sugar consumption. The World Health Organisation (WHO) recommends limiting free sugar intake to less than 10% of total daily energy consumption, with additional health benefits observed when reduced below 5% [1]. Consequently, the food industry is focusing on natural, non-caloric sweeteners to meet consumers’ demand for healthier alternatives. However, regulatory barriers significantly influence the availability and adoption of these sweeteners, shaping market dynamics and public health strategies.

Monk fruit (*Siraitia grosvenorii*), also known as Luo Han Guo, has been used in traditional Chinese medicine (TCM) for centuries for its soothing, cooling, and anti-inflammatory properties. Historical sources point to its use as early as the 13th century, mainly for treating sore throats, coughs, and digestive ailments. It owes its English name—monk fruit—to the Buddhist monks of southern China, who were the first to spread its cultivation and use [2].

MFE is rich in mogrosides—triterpene glycosides—of which mogroside V is the dominant compound, responsible for its intense sweetness, estimated to be about 250 times that of sucrose [3]. Unlike traditional sweeteners, mogrosides do not raise blood glucose levels, making monk fruit extract (MFE) a promising ingredient for dietary interventions that support metabolic health [4].

Contemporary research suggests that MFE may also exhibit other health-promoting properties beyond sweetening effects. Bioactive compounds in the extract have shown antioxidant, anti-inflammatory, hepatoprotective and antidiabetic potential in preclinical studies and emerging clinical trials [5]. Such activity supports the growing interest in monk fruit in the food industry and in integrating traditional phytotherapeutic solutions with modern nutritional strategies targeting glycaemic control and chronic disease prevention [4].

Recent systematic reviews of clinical trials have reported its beneficial effects of on glucose metabolism and insulin sensitivity. In addition to its sweetening properties, MFE shows potential as a metabolism-supporting agent. A recent systematic review of clinical trials showed that its consumption led to a reduction in postprandial glucose levels (by 10–18%) and a decrease in insulin response (by 12–22%) in healthy individuals. These effects were not associated with significant adverse effects, suggesting metabolic neutrality or benefits compared to traditional sugars. The same review also pointed to preliminary evidence of anti-inflammatory and symptom-relieving effects, which may be related to the action of mogrosides, the main bioactive compounds in the extract. In addition, a decrease in cravings for sweets and less “reward reinforcement” in response to sugar-containing products has been reported, which may support appetite control and adherence to dietary recommendations. All this makes MFE a promising ingredient in nutritional strategies aimed at glycaemic control, prevention of metabolic diseases, and long-term health support [6]. Despite these promising findings, regulatory acceptance varies considerably across global jurisdictions.

In the United States, the Food and Drug Administration (FDA) has classified MFE as Generally Recognised as Safe (GRAS), permitting its widespread use in food products [7,8]. Likewise, the National Health Commission (NHC) of China has authorised MFE as a food additive under GB 2760-2014, allowing its incorporation into a range of food categories [9]. In contrast, the European Union (EU) has historically maintained a more cautious regulatory approach. Under the EU Novel Food Regulation (EU) 2015/2283, any food not significantly consumed within the EU before May 1997 requires a comprehensive safety evaluation before market approval [10].

While the European Food Safety Authority (EFSA) issued a positive opinion in 2024 on the safety of a specific aqueous extract of MF, authorising its use under the Commission Implementing Regulation (EU) 2024/2345 [11,12,13], other MFEs, particularly highly purified mogrosides and non-aqueous extracts, remain unapproved due to gaps in toxicological data and the absence of industry-led novel food applications. It is also important to note that certain aqueous monk fruit decoctions have been recognised as non-novel foods in the UK and Ireland, permitting their use without novel food authorisation based on evidence of significant consumption before 1997 [14,15].

Although designed to protect consumers, the EU’s rigorous pre-market safety requirements can hinder innovation and delay the introduction of new food products, including natural sweeteners like MFE. In contrast, regulatory frameworks in the US and China emphasise historical use and industry-led safety assessments, enabling faster market access [8,9]. These differences have contributed to a fragmented global regulatory landscape for natural sweeteners.

Recent discussions on reforming EU food policy have suggested adopting a risk-benefit approach similar to the UK’s post-market monitoring model [14]. Such a framework could enable conditional approvals based on international safety evaluations and ongoing surveillance, balancing innovation with consumer protection. This approach was successfully implemented in the approval of steviol glycosides in 2011 following comprehensive EFSA reviews and has since facilitated the integration of other functional food components, such as oat beta-glucans and algae-derived omega-3 fatty acids, into the European market [16,17].

Despite the growing consumer interest in natural, clean-label sugar alternatives, MFE remains only partially approved in the EU, with significant regulatory hurdles persisting for highly purified forms. Addressing these challenges is essential for aligning EU food regulations with innovation goals under the European Green Deal and the Sustainable Development Goals [18]. This review analyses the regulatory barriers preventing the broader authorisation of MFE in the EU. It compares its regulatory pathway with other natural sweeteners, such as steviol glycosides and erythritol, to identify pathways for regulatory harmonisation and market integration.

## 2. Materials and Methods

This study employs a structured regulatory analysis to examine the approval status of non-caloric sweeteners in the EU and identify the regulatory barriers preventing MFE market authorisation. The study follows a three-stage analytical framework: (i) classification of authorised non-caloric sweeteners by chemical composition and regulatory status; (ii) comparative assessment of regulatory frameworks in the EU, US, and UK; and (iii) systematic extraction of regulatory data from official sources, including EU legislation, EFSA scientific opinions, FDA and FSA reports, and international food safety guidelines such as Codex Alimentarius. The analysis also considered recent regulatory developments, particularly the authorisation of aqueous MFE in the EU under Commission Implementing Regulation (EU) 2024/2345 [11,12]. This inclusion ensures that the study reflects the most current regulatory landscape as of 2025.

### 2.1. Study Design

This review follows a comparative legal and policy analysis methodology to identify key determinants influencing regulatory outcomes for MFE in the EU, based on substances currently authorised in the EU to enhance sweetness. Legal instruments and scientific evaluations were reviewed to assess regulatory differences across global jurisdictions.

### 2.2. Data Collection and Sources

Data were extracted from:EU legislation (Regulation (EU) 2015/2283 on Novel Foods; Regulation (EC) No 1333/2008 on food additives);EFSA scientific opinions on non-caloric sweeteners and novel foods;FDA and FSA regulatory reports, including Generally Recognised as Safe (GRAS) determinations and novel food dossiers;Codex Alimentarius international food safety guidelines;Commission Implementing Regulation (EU) 2024/2345 authorising aqueous monk fruit extract as a novel food;EFSA’s 2024 scientific opinion on the safety of aqueous MFE;National opinions on the novel food status of monk fruit decoctions in the UK and Ireland;UN Sustainable Development Goals documentation relevant to sustainable food systems.

Relevant policy documents were reviewed to evaluate risk assessment frameworks, precautionary principles, and the role of industry submissions in regulatory decision-making.

### 2.3. Data Analysis

Data were synthesised into regulatory comparison tables, detailing:Approval status of non-caloric sweeteners in the EU, US, and UK;Acceptable Daily Intake (ADI) values and toxicological evaluation criteria;Differences between pre-market safety assessments and post-market monitoring mechanisms.

Only documents and publications that met the following criteria were included in the review: (i) regulatory documents currently in force or published by June 2024; (ii) full-text availability in English or Polish; and (iii) direct relevance to MFE, natural sweeteners, or food-related legislative processes. Sources lacking scientific or regulatory validity, including non-peer-reviewed materials, were excluded.

## 3. Results

This regulatory analysis identified 18 non-caloric sweetening substances authorised in the EU, classified into three main categories: intense sweeteners, sugar alcohols (polyols), and sweetness modifiers. These compounds fall under Regulation (EC) No 1333/2008, which establishes permitted additives, their maximum usage levels, and the safety evaluations performed by the European Food Safety Authority (EFSA) [1]. EFSA assesses toxicological profiles, Acceptable Daily Intake (ADI) values, and potential metabolic effects before authorization [19,20].

A comparative review of MFE regulation revealed significant inconsistencies across jurisdictions. As of October 2024, the European Commission has authorised using a specific aqueous MFE as a novel food under Commission Implementing Regulation (EU) 2024/2345, permitting its incorporation into several food categories [11,12]. However, other forms of MFE, particularly highly purified mogrosides and non-aqueous extracts, remain unapproved due to gaps in toxicological data and the absence of industry-led novel food applications [11,12,13]. Notably, certain MF decoctions have been recognised as non-novel foods in the UK and Ireland, enabling their use without novel food authorization based on evidence of significant consumption before 1997 [14,15].

These findings confirm regulatory discrepancies across jurisdictions regarding monk fruit extract authorization, particularly for highly purified or non-aqueous forms.

### 3.1. Intense Sweetening Agents in the European Union

Intense sweeteners are high-potency compounds that impart sweetness without contributing significant caloric value and are widely used in beverages, sugar-free products, and low-calorie formulations. Their approval in the European Union (EU) is regulated under Regulation (EC) No 1333/2008, with periodic safety evaluations performed by the European Food Safety Authority (EFSA) to assess their toxicological profile, metabolism, and long-term health effects [1,19,20]. The regulatory characteristics, applications, safety profiles, and potential health effects of intense sweetening agents authorised for use in the European Union are summarised in Table 1.

#### 3.1.1. Acesulfame K

Acesulfame K (E950) is a heat-stable, high-intensity sweetener frequently used in soft drinks, dairy products, and confectionery. Approved under Regulation (EC) No 1333/2008, it was re-evaluated by EFSA in 2017, which confirmed its safety at an Acceptable Daily Intake (ADI) of 15 mg/kg body weight. Some studies have reported a bitter aftertaste at higher concentrations, and potential effects on gut microbiota remain under investigation, although no significant health risks have been identified within approved levels [19].

#### 3.1.2. Aspartame

Aspartame (E951) is a dipeptide-based sweetener extensively used in diet sodas, sugar-free gum, and desserts. EFSA’s 2013 comprehensive re-evaluation reaffirmed its safety, establishing an ADI of 40 mg/kg body weight. Aspartame is contraindicated for individuals with phenylketonuria (PKU), as it contains phenylalanine. No significant effects on gut microbiota have been confirmed at approved consumption levels. In addition to its role as an intense sweetener, aspartame can also function as a sweetness modifier [20].

#### 3.1.3. Cyclamates

Cyclamates (E952), widely employed in soft drinks and bakery products, have undergone several safety reassessments due to historical concerns about potential carcinogenicity, particularly bladder cancer, based on early animal studies. However, EFSA’s 2017 review confirmed that cyclamates do not pose a carcinogenic risk at approved levels, maintaining the ADI at 7 mg/kg body weight. They are authorised for use in specific food categories, including soft drinks, with maximum permitted levels of 250 mg/L [21].

#### 3.1.4. Saccharin

Saccharin (E954) is one of the oldest artificial sweeteners, primarily used in soft drinks and tabletop sweeteners. Although early studies suggested an increased risk of bladder cancer, subsequent research and EFSA evaluations have not confirmed this risk at current use levels. Saccharin is approved under Regulation (EC) No 1333/2008 with an ADI of 5 mg/kg body weight. It has a high sweetness potency but may impart a metallic aftertaste at higher concentrations [21].

#### 3.1.5. Sucralose

Sucralose (E955) is widely used in baking, beverages, and dairy products due to its high solubility and thermal stability. Approved under Regulation (EC) No 1333/2008, EFSA’s 2016 re-evaluation maintained the ADI at 5 mg/kg body weight. Although some studies have suggested potential changes in gut microbiota composition at high doses, no significant public health risks have been identified within permitted use levels [22].

#### 3.1.6. Steviol Glycosides

Derived from the Stevia rebaudiana plant, steviol glycosides (E960a) are widely accepted as natural sugar substitutes in beverages and various food products. They are thermally stable but can impart a mild liquorice-like aftertaste. EFSA’s 2010 evaluation established an ADI of 4 mg/kg body weight, confirming its safety at regulated intake levels. High intakes may lead to mild gastrointestinal discomfort in sensitive individuals [15,22].

#### 3.1.7. Neohesperidine DC

Neohesperidine dihydrochalcone (E959), derived from citrus fruit, functions as an intense sweetener and a flavour modifier due to its ability to mask bitterness. It imparts sweetness with a potential bitter aftertaste at high concentrations. EFSA’s 2022 evaluation confirmed its safety with an ADI of 20 mg/kg body weight and no significant health risks observed at approved levels [22].

#### 3.1.8. Neotame

Neotame (E961) is an intense sweetener structurally related to aspartame but substantially sweeter and used primarily in diet beverages and chewing gum. It is often combined with other sweeteners to enhance sweetness profiles. EFSA and JECFA evaluations confirmed no adverse health effects at 2 mg/kg body weight at the established ADI. Neotame has no known significant effects on gut microbiota at permitted levels [22].

#### 3.1.9. Advantame

Advantame (E969) is one of the most potent intense sweeteners, approximately 20,000 times sweeter than sucrose. It is approved under Regulation (EC) No 1333/2008 for use in baked goods, chewing gum, dairy products, and sugar-free beverages. EFSA’s 2013 assessment confirmed an ADI of 5 mg/kg body weight. Although limited metabolic data are available, no significant adverse health effects have been reported at approved usage levels [22].

### 3.2. Sugar Alcohols (Polyols) in the European Union

Sugar alcohols (polyols) are a group of sweeteners and bulk agents that contribute sweetness while providing functional properties such as moisture retention, textural stability, and reduced water activity in food products. Unlike intense sweeteners, polyols impart a moderate caloric value, typically ranging from 0.2 to 2.6 kcal/g, and generate a significantly lower glycaemic response than sucrose [23].

All polyols authorised in the European Union are regulated under Regulation (EC) No 1333/2008 on food additives. EFSA has consistently reviewed their safety, confirming that polyols are generally well tolerated when consumed within typical dietary levels. However, excessive intake can lead to gastrointestinal symptoms, including bloating, flatulence, and diarrhoea, due to their partial absorption and osmotic effects in the gastrointestinal tract [23].

Unlike intense sweeteners, polyols do not have established Acceptable Daily Intake (ADI) values, as their potential adverse effects are not toxicological but rather physiological. Regulation (EU) No 1169/2011 mandates that products containing more than 10% added polyols must carry a warning stating: “Excessive consumption may produce laxative effects” [24].

Erythritol (E968) is unique among polyols due to its high gastrointestinal tolerance. It is almost wholly absorbed in the small intestine and excreted unchanged in the urine, resulting in a significantly lower risk of gastrointestinal discomfort than other polyols [25]. Mannitol (E421) also has pharmaceutical applications as an osmotic diuretic and is used in managing elevated intracranial pressure [26].

Table 2 summarises the regulatory status, technological functions, caloric values, and safety considerations for polyols currently authorised in the European Union.

#### 3.2.1. Sorbitol

Sorbitol (E420) is a sugar alcohol derived from glucose and widely used in sugar-free confectionery, baked goods, and pharmaceutical products due to its humectant, stabilising, and texturising properties. EFSA’s 2011 opinion confirmed its safety, though excessive consumption above 20 g per day can cause gastrointestinal discomfort such as bloating and diarrhoea [23].

#### 3.2.2. Mannitol

Mannitol (E421) occurs naturally in various plants and is produced industrially by hydrogenation of fructose. It is a sweetener and stabiliser in sugar-free products and is also used pharmaceutically as an osmotic diuretic. EFSA reaffirmed its safety in 2011, noting that excessive intake (>20 g/day) may induce osmotic diarrhoea [23,26].

#### 3.2.3. Isomalt

Isomalt (E953) is derived from sucrose and is used in sugar-free confectionery and baked goods as a bulking agent and stabiliser. It does not contribute to dental caries and has a lower glycaemic impact than sucrose. EFSA’s 2017 assessment confirmed that consumption above 30 g per day may cause mild gastrointestinal discomfort, including bloating and flatulence [23].

#### 3.2.4. Maltitol

Maltitol (E965) is produced by hydrogenation of maltose and is widely used in sugar-free chocolates, baked goods, and desserts. It has a sweetness profile similar to sucrose and is non-carcinogenic. EFSA’s 2017 evaluation confirmed its safety, though excessive consumption can result in gastrointestinal discomfort [23].

#### 3.2.5. Lactitol

Lactitol (E966) is obtained from lactose and functions as a sweetener and bulking agent in sugar-free candies, baked goods, and ice cream. It has a mild sweetness and provides fewer calories than sucrose. EFSA confirmed its safety in 2017, but overconsumption can lead to bloating and diarrhoea [23].

#### 3.2.6. Xylitol

Xylitol (E967), a naturally occurring polyol in fruits and vegetables, is widely used in chewing gum, oral care products, and confectionery. It has a cooling effect and helps retain moisture, preventing food from drying. While xylitol is beneficial for dental health and non-carcinogenic, consumption above 40 g per day may produce a laxative effect and influence gut microbiota [23].

#### 3.2.7. Erythritol

Erythritol (E968) is produced by fermentation and occurs naturally in fruits such as melons and grapes. It is a low-calorie sweetener in beverages, sugar-free confectionery, and functional foods. Unlike other polyols, erythritol is fully absorbed in the small intestine and excreted unchanged, leading to minimal gastrointestinal side effects. EFSA’s 2015 evaluation confirmed its safety at typical consumption levels [23].

#### 3.2.8. Polydextrose

Polydextrose (E1200) is a randomly bonded glucose polymer used as a bulking agent and fibre source in baked goods, desserts, and dietary fibre-enriched foods. EFSA’s 2021 review affirmed its safety, although high doses may cause mild gastrointestinal discomfort and laxative effects due to incomplete digestion [23].

### 3.3. Sweetness Modifiers in the European Union

Sweetness modifiers are used to enhance the sweetness perception, to mask bitterness, or to improve the overall flavour profile of food and beverages. In contrast to intense sweeteners, primarily used to replace sugar by imparting direct sweetness, sweetness modifiers are often added in small quantities to adjust taste balance and improve palatability.

While some substances, such as thaumatin and neohesperidine dihydrochalcone, are classified purely as flavour modifiers under Regulation (EC) No 1333/2008, other compounds, such as aspartame, occupy a dual role. Although aspartame is legally classified as an intense sweetener under Annexe II of Regulation (EC) No 1333/2008, it also functions as a sweetness modifier due to its capacity to mask bitterness and enhance the taste profile of various food products. For this reason, aspartame is included in this section to reflect its significant technological role beyond simple sweetening [23].

Annexe III of Regulation (EC) No 1333/2008 further specifies labelling requirements and permitted use levels for specific sweetness modifiers. Table 3 summarises the regulatory status, typical applications, technological functions, and safety profiles of sweetness modifiers authorised in the EU.

#### 3.3.1. Thaumatin

Thaumatin (E957) is a natural sweet-tasting protein extracted from the katemfe fruit (Thaumatococcus daniellii). It functions as a sweetener and a flavour modifier by masking bitterness and enhancing flavour perception in food and beverage products. EFSA’s 2021 evaluation concluded that thaumatin does not pose toxicological risks, and no ADI has been specified due to its safe metabolic breakdown [23].

#### 3.3.2. Neohesperidine Dihydrochalcone

Neohesperidine dihydrochalcone (Neohesperidine DC; 959) is derived from bitter orange peel and functions primarily as a sweetness enhancer and bitterness masker. While it possesses significant sweetness potency, its use is mainly directed toward modifying flavour profiles rather than providing primary sweetness. EFSA’s 2022 opinion confirmed its safety at an ADI of 20 mg/kg body weight, noting no significant health risks when used within approved limits [23].

## 4. Regulatory Landscape on MFE in a Global Perspective

The regulatory framework for monk fruit extract (MFE) varies significantly across global jurisdictions, including the EU, UK, US, China, and other regions where novel food regulations continue to evolve. Comparing and understanding these differences is important to assessing the feasibility of integrating MFE into the European market and identifying potential pathways towards regulatory harmonisation.

### 4.1. European Union

In the EU, MFE is regulated under the Novel Foods Regulation (EU) 2015/2283, which requires pre-market authorisation for any food not consumed significantly in the EU before 15 May 1997 [10]. Although an aqueous MFE was authorised under Commission Implementing Regulation (EU) 2024/2345, permitting its use in specific food categories, other forms—particularly high-purity mogroside extracts—remain unapproved due to insufficient toxicological data and the absence of industry-led applications [11].

EFSA’s 2024 opinion confirmed the safety of the authorised aqueous extract within defined usage levels. However, the lack of an established Acceptable Daily Intake (ADI) for high-purity extracts further limits broader applications of MFE in the EU market. This cautious regulatory stance is designed to protect consumers but creates significant barriers for innovation and market integration of natural sweeteners such as MFE.

### 4.2. United Kingdom

Following Brexit, the UK operates an independent novel food regime overseen by the Food Standards Agency (FSA). In June 2024, the FSA recognised certain aqueous monk fruit decoctions as non-novel due to evidence of significant consumption before 1997, thus allowing their use without formal novel food authorisation in some product categories [11]. However, concentrated monk fruit extracts intended as high-intensity sweeteners remain unapproved and require novel food authorisation.

The UK’s food sector regulatory regime provides more flexibility than the EU, with conditional approvals and post-market surveillance mechanisms, which may facilitate faster adoption of innovative ingredients, including MFE [11].

### 4.3. United States

The FDA has classified MFE as Generally Recognised as Safe (GRAS), enabling both aqueous and high-purity extracts to be used broadly in food products without further pre-market approval. The GRAS system expedites market access by allowing manufacturers to rely on existing scientific data and expert consensus, provided the ingredient’s safety is sufficiently documented. As a result, MFE has become widely integrated into the US food market as a natural, calorie-free alternative to sugar, appearing in beverages, dairy products, baked goods, and other food categories [7,8].

### 4.4. China

China’s National Health Commission (NHC) has a longstanding history of regulating monk fruit as both a traditional medicine and a food ingredient. MFE is listed under GB 2760-2014 as a permitted food additive, allowing its use in various food categories, including beverages, confectionery, and health supplements [9].

China’s regulatory framework relies on the historical use of botanicals, which facilitates the approval of ingredients like MFE without requiring extensive toxicological testing that might be mandated in other regions [9]. This approach has enabled the rapid expansion of MF-based products in China’s food market.

### 4.5. Monk Fruit vs. Approved Natural Sweeteners

While natural sweeteners such as stevia and erythritol have successfully passed the EU’s Novel Food authorization process, MFE remains unapproved due to regulatory delays. Figure 1 outlines the EU approval pathway and illustrates the current status of the MFE case within this framework.

While the diagram visualises the procedural steps of the EU’s novel food approval framework and MFE’s position, the broader regulatory context also reflects differences across global markets. To complement the timeline, Table 4 provides a comparative summary of the regulatory pathways, legal grounds, and estimated authorisation timelines for monk fruit extract and two approved natural sweeteners—steviol glycosides and erythritol—in the EU, USA, and China. This comparison highlights the complexity and variability of international food regulation and its practical implications for product development and commercialisation.

The regulatory trajectory of MFE in the EU differs significantly from the more streamlined authorisation pathways of other natural sweeteners, such as steviol glycosides and erythritol. The experiences of these approved sweeteners offer critical insights into the factors facilitating or hindering regulatory progress for new natural ingredients in the EU. Table 4. provides a comparative overview of the regulatory status, legal pathways, approval timelines, and key differences for steviol glycosides, erythritol, and monk fruit extract across the European Union, United States, and China, illustrating substantial variations in regulatory outcomes and timeframes. Nonetheless, these data alone do not fully explain why monk fruit extract (MFE) remains unapproved in the EU, particularly in its purified forms. As discussed below, several factors contribute to the more complex regulatory pathway for MFE in the EU:Absence of historical consumption prior to 1997: Unlike erythritol or steviol glycosides, MFE was not significantly consumed in Europe before the cut-off date established under Regulation (EU) 2015/2283, triggering the requirement for a whole novel food authorisation process [10,12,13].Lack of industry-led applications: Steviol glycosides and erythritol benefited from proactive industry engagement, with companies submitting comprehensive safety dossiers to EFSA. In contrast, no complete application for high-purity monk fruit extracts has been submitted, hindering regulatory review [12,13,26].Complex chemical composition: MFE contains a mixture of mogrosides with varying sweetness intensities and potential biological effects, complicating toxicological assessment compared to the more chemically uniform profiles of steviol glycosides or erythritol [2,3,13].Stricter toxicological requirements: EFSA mandates extensive toxicological data, including studies on metabolism, genotoxicity, and potential effects on gut microbiota, whereas the US GRAS process and Chinese regulations place greater weight on historical use and existing scientific literature [5,6,7,9,13].Lack of established ADI: Unlike steviol glycosides (ADI 4 mg/kg bw/day) and erythritol, MFE has no established Acceptable Daily Intake in the EU, preventing its inclusion in precise food formulations and labelling [12,13,26].

These factors create a significant hurdle for MFE’s approval in the EU, contributing to delays and regulatory inertia, despite global recognition of its safety and functional benefits. This situation also impacts the competitiveness of the European food industry, which risks relying on imported finished products containing MFE rather than developing domestic innovations [9,10,11].

## 5. Discussion

This study highlights significant discrepancies in regulatory pathways for MFE between the EU and other major jurisdictions.

While MFE has been designated as GRAS in the US since 2010 [7] and has been included in China’s GB 2760 standards since 2014 [9], the EU has only recently approved a specific aqueous extract under Commission Implementing Regulation (EU) 2024/2345 [2]. However, purified extracts with high mogroside content remain unapproved due to gaps in toxicological data and a lack of new applications in the food industry [9].

These regulatory delays mirror the historical trajectory of other natural sweeteners, which required more than a decade of evaluation before gaining EU approval in 2011 [3,4].

A comparative overview of functional, metabolic, and regulatory properties of selected natural sweeteners is presented in Table 5.

The described examples underscore how comprehensive safety documentation and proactive industry involvement significantly impact the approval timeline.

Our analysis confirms that the EU regulatory system, governed by the precautionary principle in Regulation (EU) 2015/2283 [8], creates barriers to innovation in food technology leaving the sector behind other jurisdictions. As shown in Table 4, approval timelines in the EU are significantly longer than in the United States and China, where the regulatory framework allows greater reliance on historical consumption and industry-based risk assessments [7,9].

### 5.1. Policy Implications and Recommendations

The comparative analysis highlights that the EU’s regulatory approach, although grounded in consumer protection, poses significant barriers for natural sweeteners such as monk fruit extract (MFE). This differs from the more dynamic systems in the US and China, where mechanisms like the GRAS designation and established national standards enable quicker market entry [7,9].

Steviol glycosides and erythritol benefited from extensive industry engagement and comprehensive safety data, which facilitated their approvals in the EU after regulatory timelines of approximately 8 to 12 years [3,4]. In contrast, MFE lacks robust toxicological data and has no industry-led applications for purified extracts, contributing to regulatory inertia despite international acceptance [9].

This regulatory stagnation could undermine the EU’s strategic objectives under the European Green Deal and Farm to Fork Strategy, which promote healthier diets and reduced sugar consumption [11,15]. A more adaptive model—incorporating conditional approvals and reliance on international safety assessments—could improve regulatory efficiency and foster innovation, aligning with public health goals and sustainable market growth [6,8].

Therefore, policymakers should consider:Recognising credible international safety assessments, such as FDA GRAS notices, to inform EFSA evaluations;Introducing conditional approvals coupled with post-market monitoring;Providing incentives for industry stakeholders to submit novel food applications;Enhancing cooperation between the EU, UK, and global regulators to harmonise standards.

Such changes would support a competitive European food industry while maintaining high safety standards.

### 5.2. Further Research Directions

Future research should explore:Collection and analysis of proprietary industry data on toxicological studies for purified monk fruit extracts;Economic assessments to quantify the market potential for MFE in the EU and compare it with established sweeteners;Consumer acceptance studies regarding taste, safety perception, and willingness to adopt MFE as a sugar alternative.

It is important to acknowledge that this study focuses exclusively on publicly available legal and regulatory documents and does not include proprietary industry reports or detailed market data.

## 6. Conclusions

MFE is a promising next approved in UE natural sweetener with potential benefits producers and customers. However, its full regulatory acceptance in the European Union remains limited due to gaps in toxicological evidence, complex regulatory requirements, and the absence of industry-led applications for purified forms.

This regulatory situation contrasts with other jurisdictions where MFE is widely authorised and integrated into food markets. Addressing these barriers is crucial to ensure that European consumers and industry benefit equally from innovative, natural sugar alternatives.

Further dialogue and cooperation between regulators, industry stakeholders and scientific bodies will be essential to align safety standards and facilitate market access for MFE in the EU. Understanding and addressing these regulatory barriers is key to unlocking full approval of MFE in the EU and supporting innovation in natural sweeteners.

Furthermore, MFE shows significant commercial potential. According to Grand View Research, the European market for MF-based sweeteners was valued at about USD 54.7 million in 2023 and is projected to grow at a compound annual growth rate (CAGR) of 8.6% from 2024 to 2030 [27]. Additionally, the global market was valued at USD 353.7 million in 2023, forecasted to grow to USD 592.4 million by 2030 [28]. Other sources point to even higher forecasts, estimating the global market at USD 621.3 million in 2024 and USD 880.1 million in 2031 [29]. These trends reflect growing consumer demand for natural, low-glycaemic sugar substitutes in the food and beverage sector. Liberalising regulations to allow greater use of MFEs could unlock significant innovation and market potential in the European functional products industry.

## 7. Strengths of the Study

This study offers several unique strengths:**Comprehensive Legal Review:** It systematically analyses the regulatory landscape of MFE across the EU, US, UK, and China, integrating the most current legislative developments, including Regulation (EU) 2024/2345.**Practical Relevance for Industry:** The work serves as a practical compendium for food technologists and industry professionals by summarising regulatory statuses, technological functionalities, and safety profiles of both MFE and other authorised sweeteners in the EU.**Comparative Approach:** By contrasting EU procedures with faster approval pathways in the US and China, the study identifies key barriers and opportunities relevant for policy reform and market strategy.**Original Tabular Summaries:** Including detailed comparative tables offers clear, at-a-glance information not previously available in the literature, making it a valuable reference for stakeholders involved in regulatory compliance, product development, and market entry planning.**Policy-Oriented Perspective:** The analysis aligns regulatory discussion with broader public health, sustainability, and innovation goals, providing insights valuable for policymakers, industry, and scientific communities.

No prior publication has combined such a regulatory, technological, and policy-focused perspective on MFE and comparable sweeteners, filling a significant gap in the scientific and professional literature.

## Figures and Tables

**Figure 1 foods-14-02810-f001:**
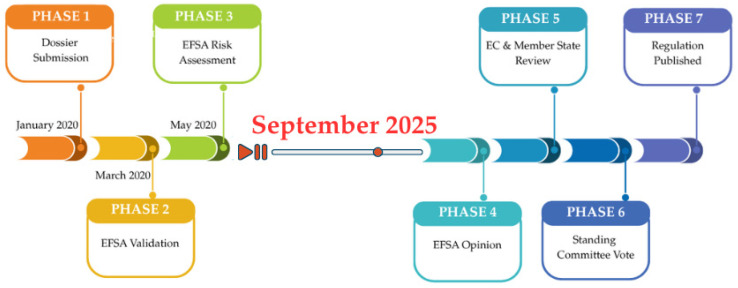
The EU Novel Food authorization pathway with current progress status for Monk Fruit Extract (MFE). As of 2024, the process remains paused at the EFSA scientific risk assessment stage due to missing toxicological data.

**Table 1 foods-14-02810-t001:** Regulatory Status and Safety Evaluation of Intense Sweetening Agents Authorised in the European Union.

Sweetener (E Number)	Common Applications	Technological Characteristics	ADI [mg/kg bw/day]	Potential Health Risks	Gut Microbiota Influence	Ref
Acesulfame K (E950)	Soft drinks, dairy products, confectionery	Heat-stable; synergistic with other sweeteners	15	Possible bitter aftertaste at high levels	Minimal effects at approved intake levels	[18]
Aspartame (E951)	Diet sodas, sugar-free gum, desserts	Dipeptide-based sweetener; high intensity	40	Contraindicated in phenylketonuria due to phenylalanine content	No significant impact at permitted levels	[19]
Cyclamates (E952)	Soft drinks, bakery products	Enhances volume; retains moisture	7	Historical concerns about bladder cancer; no risk at permitted use	Limited data; no confirmed adverse effects	[20]
Saccharin (E954)	Soft drinks, tabletop sweeteners	High sweetness; possible metallic aftertaste	5	Historical bladder cancer concerns; no risk at permitted use	Minimal impact	[21]
Sucralose (E955)	Baking, beverages, dairy products	Highly soluble; heat stable; non-caloric	5	Possible gut microbiota alterations at high doses	Potential shifts at high intake levels	[22]
Steviol Glycosides (E960a)	Natural sugar substitute, beverages	Thermally stable; mild liquorice-like taste	4	Possible mild gastrointestinal discomfort at high intake	No significant adverse effects at approved doses	[15]
Neohesperidine DC (E959)	Beverages, desserts, chewing gum	Bitter taste at high levels; flavour enhancer	20	No significant health risks observed at approved levels	Limited data	[22]
Neotame (E961)	Diet beverages, chewing gum	Synergistic with other sweeteners; minimal bulk	2	No significant health risks observed at approved levels	No known effects at permitted intake	[22]
Advantame (E969)	Baked goods, chewing gum, dairy, beverages	Ultra-high potency; heat-stable	5	No significant adverse effects reported at permitted levels	No adverse effects at approved ADI	[22]

ADI = Acceptable Daily Intake; bw = body weight. Information derived from Regulation (EC) No 1333/2008 and EFSA scientific opinions.

**Table 2 foods-14-02810-t002:** Regulatory Status, Technological Functions, and Safety Profiles of Polyols Authorised in the European Union.

Sweetener (E Number)	Common Applications	Technological Characteristics	Caloric Value [kcal/g]	Potential Health Risks	Gut Microbiota Influence	Ref.
Sorbitol (E420)	Sugar-free candies, chewing gum, baked goods	Humectant, texturizer, stabiliser	2.6	Gastrointestinal discomfort above 20 g/day	Limited impact at moderate intake	[23]
Mannitol (E421)	Sugar-free candies, pharmaceuticals	Anti-caking agent, stabiliser; medical diuretic	1.6	Osmotic diarrhoea above 20 g/day	Limited data on gut microbiota	[26]
Isomalt (E953)	Confectionery, baked goods	Bulking agent, stabiliser	2.0	Gastrointestinal discomfort above 30 g/day	Minimal impact at typical intake	[23]
Maltitol (E965)	Sugar-free chocolates,baked goods, desserts	Sweetener, texturizer	2.1	Gastrointestinal discomfort at high doses	Potential mild effects at high intake	[23]
Lactitol (E966)	Sugar-free candies, baked goods, ice cream	Sweetener, bulking agent	2.0	Bloating and diarrhoea at high intake levels	Limited data	[23]
Xylitol (E967)	Chewing gum, oral care, confectionery	Sweetener, moisture retention, cooling effect	2.4	Laxative effect above 40 g/day;potential microbiota changes	Possible shifts at high doses	[23]
Erythritol (E968)	Beverages, confectionery, functional foods	Bulk sweetener; almost fully absorbed	0.2	Generally well tolerated; minimal laxative effect	Minimal impact due to full absorption	[25]
Polydextrose (E1200)	Baked goods, desserts, fibre-enriched foods	Bulking agent, texturizer, fibre source	1.0	Gastrointestinal discomfort and mild laxative effects	Limited data	[23]

Caloric values based on EFSA assessments. Excessive intake of polyols may cause gastrointestinal discomfort. According to Regulation (EU) No 1169/2011, foods with >10% polyols require labelling: “Excessive consumption may produce laxative effects.” [14,18].

**Table 3 foods-14-02810-t003:** Regulatory Status, Technological Functions, and Safety Profiles of Intense Sweeteners and Sweetness Modifiers Authorised in the European Union.

Sweetener (E Number)	Common Applications	Technological Characteristics	ADI [mg/kg bw/day]	Potential Health Risks	Gut Microbiota Influence	Ref.
Thaumatin (E957)	Chewing gum, dairy, desserts	Natural sweet-tasting protein; flavour modifier	Not specified	No toxicological concerns at current use levels	No adverse effects reported	[23]
Neohesperidine DC (E 959)	Beverages, desserts, chewing gum	Sweetness enhancer; bitterness masking	20	No significant risks at authorised levels	Limited data available	[23]

ADI = Acceptable Daily Intake; EFSA has confirmed the safety of listed compounds within authorised use levels. For aspartame, products must carry a label indicating phenylalanine content due to health risks for individuals with phenylketonuria (PKU) [13,14].

**Table 4 foods-14-02810-t004:** Comparative Regulatory Pathways and Timelines for Steviol Glycosides, Erythritol, and Monk Fruit Extract in the EU, USA, and China.

Sweetener	Jurisdiction	Approval Status	Legal Basis/Regulatory Path	Approval Year	Approx. Time to Approval	Notes/Specifics	Ref.
Steviol Glycosides (Stevia)	EU	Approved	Reg. (EC) No 1333/2008	2011	~10–12 years	Industry-led dossier; EFSA safety review	[1,3]
	USA	GRAS	GRAS self-notification	2008	<1 year	Widely used across food categories	[7]
	China	Approved	GB 2760 (1996)	1996	<1 year	Extensive historical use	[9]
Erythritol	EU	Approved	Reg. (EC) No 1333/2008	2008	~8–10 years	Industry-led dossier; EFSA safety review	[1,4]
	USA	GRAS	GRAS self-notification	2001	<1 year	Non-cariogenic, low-calorie polyol	[8]
	China	Approved	GB 2760 (~1997)	~1997	<1 year	Widely used in confectionery	[9]
Monk Fruit Extract (Aqueous)	EU	Approved (limited aqueous form)	Reg. (EU) 2015/2283	2024	No prior applications	Only specific aqueous decoction allowed	[2,5,6]
	USA	GRAS	GRAS self-notification	2010	<1 year	Covers both aqueous and purified forms	[7]
	China	Approved	GB 2760 (2014)	2014	<1 year	Permitted as food additive	[9]
Monk Fruit Extract (Purified)	EU	Not Approved	Reg. (EU) 2015/2283	—	No application submitted	Requires new safety dossier	[2,5,6]
	USA	GRAS	GRAS self-notification	2010	<1 year	Permitted in multiple applications	[7]
	China	Approved	GB 2760 (2014)	2014	<1 year	No specific restrictions for purity	[9]

**Table 5 foods-14-02810-t005:** Comparison of functional, metabolic, and regulatory properties of selected natural sweeteners.

Property	MFE	Stevia	Erythritol	Xylitol	References
Sweetness intensity	~250×	~200–300×	~0.6–0.8×	~1×	[3]
Caloric content (kcal/g)	0	0	~0.2	~2.4	[3]
Glycaemic impact	Neutral effect on blood glucose	Neutral	Neutral	Slight glycaemic increase	[4]
Biological activity	Antioxidant, anti-inflammatory, hepatoprotective	Antioxidant, potentially antihypertensive	Antioxidant, prebiotic	Antibacterial (oral cavity), prebiotic	[5]
Digestive tolerance	High (non-fermentable)	High	High in moderate doses	May cause laxative effect in higher amounts	[3]
Legal status in EU	Not approved	Approved	Approved	Approved	[3]
Source	Fruit of *Siraitia grosvenorii*	Leaves of *Stevia rebaudiana*	Naturally in fruits, industrially from glucose	Naturally in fruits/vegetables, industrial production from hemicellulose	[2,3]

Sweetness intensity is expressed relative to sucrose (e.g., 250× = 250 times sweeter than sucrose). Caloric values refer to average energy density per gram. “Neutral glycaemic effect” denotes no significant rise in blood glucose or insulin response after consumption. Legal status refers to the European Union regulatory framework as of 2024.

## Data Availability

The original contributions presented in the study are included in the article. Further inquiries can be directed to the corresponding author.

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
