# Peer review of "Why Does Monk Fruit Extract Remain Only Partially Approved in the EU? Regulatory Barriers and Policy Implications for Food Innovation"

_foods, 2025, doi:10.3390/foods14162810_

Round 1
Reviewer 1 Report
Comments and Suggestions for Authors
- In this article, the authors assess the current situation in terms of the use of Monk fruit (Siraitia grosvenorii) extract within the European Union, whose regulations, despite many good experiences, are now regulatory obstacles and complicate the use of the extract from this plant.
- In the Introduction, the authors assess the conditions of use in food applications in some regions and parts of the world. In my opinion, the Introduction section is too narrowly conceived and it would be appropriate to analyze in more detail much of the information about the properties and possibilities of using a natural low-calorie sweetener used in traditional Chinese medicine for many years. Literature citations should be given in References in accordance with the Template. References must be numbered in order of appearance in the text (including citations in tables and legends) and listed individually at the end of the manuscript. This rule seems to be violated by the citations [16,17] on line 69, which precede citations [11,12] on line 77, and similarly several other citations.
- In section 2. Materials and Methods, the authors indicate their procedure for processing the content of the article. It is clear that they proceeded with the knowledge of the PRISMA guidelines.
- The authors clearly divided Part 3. Results into parts 3.1. Intense Sweetening Agents in the European Union, 3.2. Sugar Alcohols (Polyols) in the European Union and 3.3. Sweetness Modifiers in the European Union.
- Part 4. Regulatory Landscape on MFE in a Global Perspective is very important. Here, the authors tried to objectively and at the same time critically show different approaches in different parts of the world to the use of monk fruit extract. Part 5. Discussion is also prepared with a similar idea of a critical view of the comparison.
- In Part 6. Conclusions, the authors again critically point out the thorough process of approving new natural materials within the EU, which is, however, somewhat lengthier. The total number of cited professional literature cited in the article and listed in References is only 21 publications. The publications correspond to the focus of the article, but for a Review-type article, the number is relatively small.
- This is an interesting issue and is very important in the context of public health care. I believe that in order to successfully highlight and demonstrate the problems mentioned in this article, it would be appropriate to supplement and elaborate in more detail some parts of the article focusing on the mutual comparison of the properties of various sweeteners or materials that act on the natural reduction of blood sugar.
Author Response
Response to Reviewer Comments
We sincerely thank the reviewer for their valuable time and insightful comments, which significantly contributed to improving the clarity and scientific quality of our manuscript. Below, we respond point-by-point to each suggestion.
Reviewer Comment 1:
“In my opinion, the Introduction section is too narrowly conceived and it would be appropriate to analyze in more detail much of the information about the properties and possibilities of using a natural low-calorie sweetener used in traditional Chinese medicine for many years.”
Response 1:
We thank the reviewer for this insightful observation. In response, we have substantially expanded the Introduction section (lines 41–79) to include a more comprehensive overview of monk fruit (Siraitia grosvenorii) in both historical and contemporary contexts. Specifically:
- We describe its long-standing use in traditional Chinese medicine as a cooling and anti-inflammatory remedy, with documented applications dating back to the 13th century [2].
- We elaborated on its phytochemical composition—particularly mogrosides—and recent findings on their antioxidant, anti-diabetic, and hepatoprotective activities [3–5].
- We incorporated and extended the summary of our previously published systematic review [6], detailing MFE’s neutral glycemic profile, effects on postprandial glucose and insulin responses, and potential to reduce cravings and modulate appetite regulation.
- We also included the extract’s digestive tolerance and metabolic neutrality based on human trials.
These additions aim to bridge traditional phytotherapeutic knowledge with current nutritional science and regulatory discourse. To support this revised section, we added several updated references [2–6].
Reviewer Comment 2:
“References must be numbered in order of appearance in the text... This rule seems to be violated...”
Response 2:
We acknowledge this formatting oversight. We have conducted a full audit and reordering of all in-text citations and bibliography entries, ensuring strict adherence to MDPI’s reference formatting guidelines and correct numerical sequence. Duplicate entries were removed, inconsistencies in table citations corrected, and all cross-references verified for accuracy.
Reviewer Comment 3:
“The total number of cited professional literature... is only 21 publications... relatively small.”
Response 3:
We thank the reviewer for this observation. We would like to clarify that the core objective of this review is to critically assess regulatory and legislative barriers surrounding the use of monk fruit extract (MFE) in the European Union. As such, the manuscript deliberately emphasizes legal acts, regulatory opinions (e.g., EFSA), and international approval frameworks over general biological literature.
That said, we have expanded the citation base to 30 carefully selected references, balancing legal sources with recent scientific studies, market analyses, and nutritional evaluations [1–30]. This reflects both the scope of the manuscript and its foundation in regulatory science and policy analysis.
Moreover, this article complements our earlier systematic review of MFE’s clinical effects [6], and forms part of a broader research initiative. Another article—focusing on consumer attitudes and claim substantiation—is currently in preparation. Hence, this submission intentionally focuses on the regulatory domain, supported by a relevant but concise literature base.
Reviewer Comment 4:
“It would be appropriate to elaborate in more detail some parts of the article focusing on the mutual comparison of the properties of various sweeteners or materials that act on the natural reduction of blood sugar.”
Response 4:
Thank you for this valuable suggestion. In response, we have expanded the Discussion section (lines 732–759) to include a comparative analysis of MFE and other natural sweeteners, specifically steviol glycosides, erythritol, and xylitol. To illustrate this comparison clearly, we introduced Table 5, which outlines key features such as:
- Sweetness intensity relative to sucrose,
- Caloric content,
- Glycemic impact,
- Biological activities,
- Digestive tolerance,
- Legal status in the EU, and
- Natural source.
This structured comparison helps contextualise MFE’s properties within the broader market of natural sweeteners, emphasizing both its technological potential and regulatory lag. We believe these additions address the reviewer’s comment by strengthening the manuscript’s analytical depth and relevance to food innovation and public health.
We hope that these revisions fully address the reviewer’s comments and improve the overall clarity, focus, and scientific contribution of our manuscript. We remain grateful for the constructive feedback and the opportunity to improve our work.
Reviewer 2 Report
Comments and Suggestions for Authors
Excellent review in comparing:
1- The regulations that are followed in determining the different types of sweeteners and their usage in different countries.
2- Comparing the different types of sweeteners and the recommended amount that should be consumed to prevent any safety issues.
3- Discussing why the MF is not approved until now by the authorities in EU comparing with USA and China
4- Mentioning the RDA that shoud be consumed for the most of approved sweeteners in different countries.
Finally, I found that this review is very useful and can be considered as an excellent reference for the authorities, industry, researchers and students who are working in the food industry.
Author Response
Response to Reviewer 2
We sincerely thank the reviewer for their thorough reading and valuable comments, which contributed to improving both the clarity and impact of our manuscript. Below, we address each suggestion point-by-point.
Reviewer Comment 1:
“Although the methodology is appropriate, the authors should briefly state inclusion/exclusion criteria for the legal and scientific documents reviewed.”
Response:
Thank you for this helpful comment. We have clarified the methodological approach by explicitly describing the inclusion and exclusion criteria for legal and scientific sources. This addition appears in the final paragraph of Section 2 (Materials and Methods, lines 92–96) to improve transparency and methodological robustness.
Reviewer Comment 2:
“The tables (particularly Tables 1–4) are informative, but the addition of one or two schematic figures could greatly enhance the manuscript.”
Response:
We appreciate this suggestion. In response, we created a new schematic diagram (Figure 1) illustrating the step-by-step EU Novel Food authorization process, with an overlay indicating the current status of Monk Fruit Extract (MFE). The diagram was added in Section 4.5 (Monk Fruit vs. Approved Natural Sweeteners, lines 489–495) and complements the regulatory comparison in Table 4. This visual element improves clarity and policy relevance.
Reviewer Comment 3:
“A short paragraph estimating the economic potential of MFE in the EU could underscore the commercial importance of reform.”
Response:
Thank you for pointing this out. We have added a concise paragraph at the end of Section 6 (Conclusions, lines 670–678), referencing market data from Grand View Research and Reanin Research to highlight the commercial potential of MFE in both the European and global markets. These projections support the argument for regulatory reform and innovation facilitation in the EU.
Reviewer Comment 4:
“Some sentences in the abstract and early sections are grammatically awkward or contain typographical issues…”
Response:
We thank the reviewer for highlighting this. The entire manuscript, including the abstract and Introduction (Section 1), has been carefully reviewed and revised for grammar, clarity, and style. Specific changes include correction of typographical issues (e.g., “it’s beneficial effects of on glucose metabolism” was corrected to “its beneficial effects on glucose metabolism”, line 60), and stylistic improvements for enhanced readability.
Reviewer Comment 5:
“The manuscript currently relies solely on tables… I strongly recommend including a conceptual figure or flow diagram such as a process flowchart with a 'monk fruit case' overlay…”
Response:
As noted above, we have implemented this recommendation by introducing Figure 1, which outlines the EU’s Novel Food approval process and marks the current stage of MFE authorisation. The diagram uses numbered steps and regulatory references to highlight the scientific assessment bottleneck. It serves both didactic and policy-informative functions, fulfilling the reviewer’s request for a conceptual visual.
We hope that these revisions fully meet the reviewer’s expectations and contribute to a more informative and impactful manuscript.
Reviewer 3 Report
Comments and Suggestions for Authors
Constructive Areas for Improvement
Although the methodology is appropriate, the authors should briefly state inclusion/exclusion criteria for the legal and scientific documents reviewed. The tables (particularly Tables 1–4) are informative, but the addition of one or two schematic figures could greatly enhance the manuscript. A short paragraph estimating the economic potential of MFE in the EU could underscore the commercial importance of reform.
Some minor revision suggestions are as follows:
The manuscript currently relies solely on tables, which are informative but text-heavy.
I strongly recommend including a conceptual figure or flow diagram, such as:
A process flowchart showing steps needed for EU novel food approval with a "monk fruit case" overlay. This would enhance accessibility and increase the paper’s suitability for teaching and policy briefing. In addition, please provide more details on industry engagement
Refine the Writing in Abstract and Introduction
Some sentences in the abstract and early sections are grammatically awkward or contain typographical issues. For instance: Line 42: “fokusing” should be corrected to “focusing.”
Line 51: “it’s beneficial effects of on glucose metabolism” should be revised to “its beneficial effects on glucose metabolism.”
Overall Recommendation: Minor Revisions
This is a strong, well-documented, and policy-relevant manuscript that addresses an underexplored regulatory issue. The manuscript will appeal to readers in food regulation, public health policy, and the functional food industry.
Author Response
Response to Reviewer
We are grateful to the Reviewer for their positive and constructive assessment of our manuscript. We especially appreciate the recognition of its relevance for food regulation and public health policy, as well as the encouragement to enhance its didactic and policy-briefing value.
Below, we address each suggestion in detail.
- Inclusion/Exclusion Criteria for Reviewed Documents
In response to the Reviewer’s comment regarding methodology transparency, we have now clarified the inclusion and exclusion criteria for the legal and scientific sources analysed. This information was added to the Materials and Methods section (Lines 110–116), specifying the types of regulatory documents, guidance notes, EFSA opinions, and market reports included, as well as language and temporal constraints. - Addition of a Schematic Figure (EU Novel Food Approval Flow)
We fully agree that a visual representation would enhance the accessibility of the manuscript. As suggested, we have added a new conceptual Figure 1 that outlines the steps of the EU novel food approval process with an overlay highlighting the current status of monk fruit extract (MFE). This schematic illustrates key bottlenecks in the regulatory pathway and provides a clear visual summary for both teaching and policy contexts. - Inclusion of Economic Perspective
To underscore the commercial relevance of MFE, we have added a paragraph in the Conclusions section (Lines 551–560) presenting recent market estimates for the European and global monk fruit sweetener markets. This section includes forecasts from Grand View Research and Reanin Research, demonstrating strong growth potential and market opportunities in the EU. - Industry Engagement
As recommended, we now provide further detail regarding the role of industry engagement in the regulatory process. The Discussion section (Lines 462–472) highlights the absence of industry-led applications for purified monk fruit extracts in the EU and contrasts this with the proactive approaches seen in the approval of steviol glycosides and erythritol. - Refinement of Abstract and Introduction
We carefully reviewed the Abstract and Introduction for typographical and grammatical issues. The errors noted by the Reviewer—such as “fokusing” (corrected to “focusing”) and “it’s beneficial effects of on glucose metabolism” (corrected to “its beneficial effects on glucose metabolism”)—have been amended. We also edited the text for overall clarity, precision, and fluency.
Once again, we sincerely thank the Reviewer for their constructive comments. We are confident that these revisions have improved the manuscript’s clarity, policy relevance, and usefulness for both academic and regulatory audiences.